# Multimodality Cardiac Imaging in Cardiomyopathies: From Diagnosis to Prognosis

**DOI:** 10.3390/jcm11030578

**Published:** 2022-01-24

**Authors:** Guillem Casas, José F. Rodríguez-Palomares

**Affiliations:** 1Cardiovascular Imaging Unit and Inherited Cardiovascular Diseases Unit, Cardiology Department, Hospital Universitari Vall d’Hebron, Vall d’Hebron Institut de Recerca, 08035 Barcelona, Spain; 2Department de Medicina, Universitat Autónoma de Barcelona, 08035 Barcelona, Spain; 3Centro de Investigación Biomédica en Red en Enfermedades Cardiovasculares, 28029 Madrid, Spain

**Keywords:** cardiomyopathy, multimodality cardiac imaging techniques, diagnosis, prognosis, left ventricular ejection fraction, late gadolinium enhancement

## Abstract

Cardiomyopathies are a group of structural and/or functional myocardial disorders which encompasses hypertrophic, dilated, arrhythmogenic, restrictive, and other cardiomyopathies. Multimodality cardiac imaging techniques are the cornerstone of cardiomyopathy diagnosis; transthoracic echocardiography should be the first-line imaging modality due to its availability, and diagnosis should be confirmed by cardiovascular magnetic resonance, which will provide more accurate morphologic and functional information, as well as extensive tissue characterization. Multimodality cardiac imaging techniques are also essential in assessing the prognosis of patients with cardiomyopathies; left ventricular ejection fraction and late gadolinium enhancement are two of the main variables used for risk stratification, and they are incorporated into clinical practice guidelines. Finally, periodic testing with cardiac imaging techniques should also be performed due to the evolving and progressive natural history of most cardiomyopathies.

## 1. Introduction

Cardiomyopathies constitute a heterogeneous group of diseases that affect the muscle of the heart and present a very diverse etiology. Classically, the European Society of Cardiology (ESC) classifies these diseases as hypertrophic, dilated, arrhythmogenic, restrictive, or other cardiomyopathies [1]. Additionally, all of them are subclassified as familial/genetic or non-familial/non-genetic. Cardiomyopathies present variable expressions and symptoms that can change over time. Thus, periodic evaluation using cardiac imaging techniques is essential throughout follow-up. These techniques can help us to diagnose, guide treatment, and optimize patients’ prognosis. 

The patient evaluation includes anamnesis, physical examination, an electrocardiogram, and transthoracic echocardiography (TTE) that may raise suspicion of cardiomyopathy. This information is usually complemented with a cardiovascular magnetic resonance (CMR) that provides more precise anatomic and functional evaluation, as well as excellent tissue characterization with prognostic implications. Additionally, some patients may require a nuclear medicine test or cardiovascular computed tomography (CT).

In this manuscript, we will review the information that the different imaging techniques offer in the diagnosis and management of these patients. 

## 2. Hypertrophic Cardiomyopathy 

Hypertrophic cardiomyopathy (HCM) is defined by an increase in left ventricular (LV) wall thickness and/or LV mass, unexplained by LV loading conditions. HCM is the most prevalent cardiomyopathy, affecting approximately 1:500 of the adult population, and is usually caused by mutations in sarcomeric genes [2,3]. 

TTE is the first-line imaging modality for HCM evaluation: a maximal wall thickness >15 mm (or higher than two standard deviations from normal corrected for age, gender, and height) or asymmetric septal hypertrophy (septal/posterior wall thickness) ratio >1.3 (or >1.5 in hypertensive patients) is suggestive of HCM (Figure 1) [4]. A wall thickness ≥30 mm is associated with a higher risk of sudden cardiac death (SCD) [2,3]. 

TTE allows for localization of the hypertrophy and hence identification of different phenotypes: septal, septal reverse, apical, or diffuse HCM, among others (Figure 2). In addition, patients with HCM present ancillary signs that, although non-specific, can help in the diagnosis: papillary muscle abnormalities (hypertrophic, bifid or trifid, and with an apical insertion), false tendons, myocardial clefts or crypts, aneurysms, or mitral valve and subvalvular structure abnormalities (elongated mitral leaflets with/without systolic anterior motion -SAM-) (Figure 3) [5]. 

Moreover, the presence of LV outflow tract obstruction (LVOTO) can be easily evaluated and is defined as a peak gradient higher than 30 mm Hg at rest or after provocative maneuvers (typically observed as a dagger-shaped curve on continuous wave Doppler, Figure 3D). LVOTO differentiates between obstructive and non-obstructive HCM, with important therapeutic and prognostic implications [2,3]. Assessment of LV systolic function or LV ejection fraction (LVEF) is also important because it remains an important predictor of events [6]. Patients with an LVEF <50% present a poor prognosis [7] and should be considered for prophylactic implantable cardioverter-defibrillator (ICD) according to the recent American guidelines [3]. In addition, the anteroposterior left atrial diameter should be measured, and the diastolic function should be assessed. The maximal wall thickness, the left atrial diameter, and LVOTO have been incorporated into the risk prediction model of SCD [8] endorsed by the ESC [2]. 

Strain parameters from speckle-tracking TTE constitute markers of LV systolic function and are more sensitive than LVEF to systolic impairment. An abnormal global longitudinal strain (GLS) has been associated with worse outcomes in HCM [9], even though clear cut-off values are not available due to lack of standardization. Other techniques, such as circumferential strain or LV rotation and twist mechanics, are not routinely recommended.

CMR has become the gold standard technique for HCM evaluation and should be performed in all patients [4]. Beyond more accurate measurement of LV wall thickness and LVEF, CMR allows for tissue characterization with late gadolinium enhancement (LGE) sequences (Figure 4). Artificial intelligence and machine learning could further improve the accuracy of wall thickness measurements [10].

LGE is a frequent finding in HCM (about 60% of patients) [6,11,12,13], correlates with the presence of myocardial fibrosis [14], and usually affects the most hypertrophied segments with an intramural pattern. In addition, its location in areas of septal-free wall junctions is a common finding. The presence of LGE has been consistently associated with adverse outcomes and especially with SCD risk [11,15,16]. 

Quantification of LGE, usually as the percentage of total LV mass using the five standard deviation method [17], has emerged as a powerful predictor of SCD, both in obstructive and non-obstructive, low- and high-risk HCM [6,11,12,13,18] and with better predictive value than the ESC calculator [8]. An LGE >15% has been proposed as a risk marker for SCD [3,16,19]. Contrarily, outcomes of patients with a low LGE amount (<5%) are comparable to those without LGE [6,12]. Progression of LGE extension throughout follow-up has also been reported in HCM patients and has been related to adverse cardiac remodeling and worse prognosis [18]. Owing to the strong evidence, “extensive” LGE has been incorporated as a parameter for prophylactic ICD implantation in the recent American guidelines, which also recommend performing a CMR every 3–5 years to evaluate disease progression [3].

Emerging CMR sequences, such as T1 and T2 mapping and extracellular volume (ECV), allow for quantitative analysis of tissue characteristics and are more sensitive than LGE for the detection of myocardial fibrosis, especially in case of diffuse interstitial fibrosis (Figure 5) [20]. Further, T1 and T2 mapping are useful in the differential diagnosis of left ventricular hypertrophy [21]. Small studies have so far reported prognostic implications of T1 mapping [22] and ECV [23] values. However, further studies are needed to validate the prognostic implications of mapping sequences to fully incorporate them into daily clinical practice.

Stress echocardiography (SE) is another useful technique in the assessment of HCM because one-third of patients have latent LVOTO. SE is safe and usually performed in a semi-supine position, while the use of dobutamine is not recommended in HCM. LVOTO, LV systolic function (LVEF and GLS), LV diastolic function (E/E’), dynamic mitral regurgitation (usually secondary to SAM) and tricuspid regurgitation (TR) velocity should be evaluated at rest, at peak stress, and post-exercise (Figure 6) [24]. Exercise-induced significant LVOTO (>50 mmHg) is a marker of worse prognosis and can be used to guide treatment [2,3,24]. A diastolic stress test may also be performed; an increase in E/E’ > 14 and peak TR velocity >2.8 m/s has been invasively correlated with elevated LV filling pressures during exercise [25] and is a marker of poor exercise tolerance [24]. The role of SE in the detection of inducible ischaemia in HCM is controversial. In this scenario, stress CMR with vasodilators is usually preferred, and signs of microvascular dysfunction may be observed [4]. 

The role of cardiac CT in HCM is secondary and may be considered to exclude coronary artery disease. Recently, assessment of myocardial fibrosis by delayed enhanced CT has been described with an adequate agreement with LGE by CMR [26]. Even though data is still scarce, and no clinical implications should be attributed, it might become an alternative for those patients who have a contraindication for CMR.

Table 1 describes the main imaging diagnostic and prognostic findings in HCM.

## 3. Dilated Cardiomyopathy

Dilated cardiomyopathy (DCM) is defined by LV (or biventricular) systolic dysfunction (LVEF < 45%) and dilatation (LV end-diastolic volumes or diameters >2 standard deviations from normal corrected by age, gender, and body surface area) not attributed to loading conditions or coronary artery disease (Figure 7). DCM affects approximately 1:2500 adults, can be both genetic and acquired, and is a leading cause of heart failure [27].

TTE allows for both anatomical and functional assessment. Three-dimensional transthoracic echocardiography (3D TTE) is increasingly used due to improvements in technology and automated software and has the advantage of direct LV measurements with no geometric assumptions (Figure 8). Most importantly, 3D TTE is more accurate and reproducible than conventional two-dimensional (2D) TTE and shows better agreement with CMR values. However, 3D TTE depends on good image quality, requires more advanced technical skills, and normal values are less well established [28].

LVEF remains the strongest predictor of events in DCM; thus, patients with an LVEF ≤35% must be considered for optimal medical treatment and prophylactic ICD (with/without cardiac resynchronization therapy, CRT) [29,30]. However, an LVEF ≤35% has proven to have a low sensibility and specificity for SCD prediction, so additional predictors are required. Strain parameters, both GLS (Figure 9) and global circumferential strain (GCS) have been associated with cardiovascular events and present an incremental prognostic value over LVEF [31]. Myocardial work (MW) is a novel quantitative parameter that incorporates both strain and LV pressure variables (Figure 10) [32]. Initial reports suggest that not only MW is related to outcomes, but also has an additional prognostic role over both LVEF and GLS in DCM [33]. 

Stress echocardiography (SE) is also useful for risk stratification in DCM. Dobutamine is most often used, with either low or high-dose protocols (10 or 40 μg/kg/min), and the contractile response is assessed. An increase by ≥5% in LVEF at peak stress is defined as a contractile reserve and is associated with improved outcomes and reverse remodeling in DCM [24,34]. SE may also be used to exclude ischaemic etiology.

CMR should be performed at least once in all DCM patients since it is the gold standard technique for the assessment of biventricular volumes, systolic function, and tissue characterization. LGE is common in DCM and traduces myocardial fibrosis [35,36], usually with a septal mid-wall pattern (Figure 11). LGE has been consistently described as a strong and independent predictor of outcomes in DCM [35,36,37], and specifically for SCD [35,37,38]. The absence of LGE is also associated with reverse remodeling [37], which confers improved survival in DCM. Of note, LGE shows incremental prognostic value over LVEF [39], even in patients with only mild or moderate systolic dysfunction [40]. 

The extension, localization, and pattern of LGE have also been associated with prognosis [41,42]; a higher burden of LGE, septal LGE, and multiple patterns (combined septal and free-wall, epicardial, and transmural LGE) are related to higher mortality or SCD. The risk of adverse outcomes is also higher in patients with progression of LGE over time [43]. Considering the important prognostic role of LGE and the aforementioned limitations of LVEF, a combined algorithm has recently been proposed to enhance the identification of patients at high risk for SCD who should be considered for prophylactic ICD [42]. However, clinical guidelines are only based on LVEF [29,30].

GLS by CMR (Figure 12) has been shown to be associated with outcomes in DCM, and most importantly to improve risk classification beyond LVEF and LGE [44]. However, it has not been standardized and no cut-off points have been proposed, so routine clinical use is not yet recommended. Mapping techniques have also been evaluated in DCM; higher T1 and ECV values have been shown to have prognostic implications irrespective of LVEF and LGE [45]. Furthermore, an increased native T2 value indicates the presence of myocardial edema, which could suggest the presence of inflammatory cardiomyopathy [46]. These techniques offer promising new tools for risk stratification, but further validation is still required.

Assessment of right ventricle (RV) size and systolic function is recommended in all patients since RV systolic dysfunction is a common finding and an independent predictor of poor outcome [47]. Different techniques may be applied: fractional area change [48], peak longitudinal strain of RV free wall [49] by TTE, or RV ejection fraction (RVEF) by CMR [47]. The same advantages and limitations previously mentioned in 3D TTE also apply to the RV; specifically, global chamber including RV inflow and outflow tracts and apical regions can only be assessed by 3D TTE and not 2D TTE (Figure 13) [28]. Secondary (functional) mitral regurgitation and diastolic dysfunction should also be evaluated since they are associated with cardiovascular events [50]. LV GLS [51], left atrial strain [52] and MW [53,54] by TTE, as well as septal viability on CMR [54], have been identified as predictors of response to CRT. 

Other imaging modalities are rarely used in DCM. Nuclear imaging can analyze cardiac sympathetic innervation, which has been associated with ventricular arrhythmias and adverse prognosis [55]. Cardiac CT is commonly used to exclude coronary artery disease. Recent studies have shown the ability of delayed enhancement CT to detect myocardial fibrosis, with comparable performance to LGE by CMR [56]. However, results are still preliminary, and no clinical studies are available, so routine use is not recommended.

Table 2 describes the main imaging prognostic findings in DCM.

## 4. Restrictive Cardiomyopathies

Restrictive cardiomyopathies (RCM) account for less than 5% of all cardiomyopathies and have a highly varied etiology (Table 3).

RCM is characterized by a marked alteration of myocardial compliance: severe diastolic dysfunction and a preserved systolic function (at least in early stages). The initial diagnosis is performed by TTE showing a normal/increased LV wall thickness (generally with a concentric/symmetric distribution), a restrictive pattern by Doppler, absence of left ventricular dilatation, preserved LVEF, and a marked biatrial dilatation [57]. However, although TTE is crucial for the initial approach and raising diagnostic suspicions, its role is limited when establishing the differential diagnosis, in which case CMR is highly relevant. 

### 4.1. Idiopathic Restrictive Cardiomyopathy

Idiopathic restrictive cardiomyopathy is a very uncommon disease affecting predominantly children and young adults with a familial pattern. It is characterized by the presence of a restrictive diastolic filling pattern (increased left ventricular end-diastolic pressure), normal left ventricular dimensions, absence of an increased left ventricular mass, normal left and right ventricular function, and absence of any other cardiac or pericardial diseases [58]. 

### 4.2. Cardiac Amyloidosis

Cardiac amyloidosis (CA) is an infiltrative disease caused by an extracellular accumulation of amyloid fibers. Most patients are affected by light chains (primary or AL amyloidosis) or by transthyretin (ATTR amyloidosis), either the hereditary form (ATTRm) or the wild type (ATTRwt) [59]. 

Among the main TTE findings, the following “red flags” should raise suspicion of CA [60]: left ventricular wall thickness > 12 mm, myocardial sparkling (Figure 14), increased valvular thickness, thick interatrial septum, low stroke volume, paradoxical low-flow low-gradient aortic stenosis, restrictive filling pattern by Doppler, and apical sparing pattern on strain analysis (a reduced global longitudinal strain with an apical to basal deformation ratio >2.1) (Figure 14), abnormal left atrial strain and pleural or pericardial effusion. 

The presence of a reduced GLS presents prognostic implications; a GLS >−14.8% has been associated with an increase in global mortality [61]. Although the apical sparing pattern is quite suggestive of CA, this pattern may not be present in patients with concomitant significant aortic stenosis [62].

CMR provides high-resolution structural and functional information, allows tissue characterization [60], and permits diagnosis at early stages compared to TTE [63]. The main findings, in addition to those described by TTE, are the difficulty to null the myocardial signal in LGE sequences, a global or diffuse LGE, and a marked increase in native T1 and ECV values (>40%) (Figure 15) [60,63,64].

As the disease progresses, there is an increase in LGE uptake, which is initially subendocardial and progressively becomes transmural (which is associated with an increase in overall mortality regardless of the type of amyloidosis) [65]. Additionally, a recent meta-analysis showed that ECV is the strongest diagnostic and prognostic imaging biomarker in CA [66], and an ECV >58% is associated with increased mortality in ATTR [67]. CMR cannot distinguish ATTR and AL forms, but there are characteristically-associated signs (Table 4) [64,65,67,68].

Either 99m technetium diphosphonate (Tc-DPD) or pyrophosphate (Tc-PYP) scintigraphy play a relevant role in diagnosing ATTR amyloidosis and allow early detection of cardiac involvement before TTE and CMR, however, false positives can be present [69]. Results are evaluated based on the Perugini scale: a grade 2 or 3 uptake (cardiac uptake similar to or greater than the ribs with/without a reduction in the bone uptake) has a positive predictive value around 100% [70]. Another means of quantification is the assessment of cardiac uptake compared to the contralateral chest (C/CL): a ratio ≥1.5 suggests the diagnosis of ATTR [68,70]. Prognostic implications of scintigraphy are still limited, however, a C/CL uptake >1.6 [71] or an apical sparing pattern [72] are associated with lower survival.

Table 5 describes the main imaging findings that should raise suspicion of cardiac amyloidosis. 

### 4.3. Fabry Disease

Anderson–Fabry disease (FD) is a rare (approximate incidence 1:40,000) genetic lysosomal storage disorder caused by a mutation in the alpha-galactosidase A (GLA) gene with an X-linked inheritance.

The presence of concentric left-ventricular hypertrophy (LVH) is the most common finding in FD, however, other patterns have also been described: septal, asymmetric, or apical [73]. LVH is higher in men than in women, and it tends to appear at younger ages. Another common finding is the presence of a binary septum: a hyperechoic endocardium adjacent to a hypoechoic subendocardium (Figure 16A). Prominent papillary muscles have also been described, as well as RV hypertrophy. There is also dilation and left atrial dysfunction (systolic and early diastolic strain) that is associated with a higher incidence of supraventricular arrhythmias. Mitral and aortic valvular thickening are also frequent. 

Generally, the biventricular systolic function is preserved until advanced stages of the disease, at least in terms of LVEF. However, FD patients show reduced values of GLS and GCS compared to controls, which are more reduced in those with inferolateral LGE (Figure 16C); thus, an inferolateral longitudinal strain peak value <−12.5% suggests the presence of fibrosis with a sensitivity of 90% and a specificity of 97% [74].

Table 6 describes the main imaging findings in Fabry disease.

The most common finding in CMR is also the presence of concentric LVH. The typical LGE pattern is an intramyocardial uptake at the basal LV inferolateral wall, which is present in up to 50% of patients [75], and may progress to transmural extension together with a marked thinning of the myocardial wall (Figure 17). 

The presence of LGE is indicative of irreversible myocardial damage; tfhus, clinical practice guidelines recommend starting treatment with a class I indication when there is no or minimal fibrosis [76]. The presence of LGE is associated with a poorer response to medical treatment and an increased risk of cardiovascular events including SCD [74]. For this reason, some authors have suggested an ICD implantation in patients with a significant LGE mass; however, there are no available guidelines. 

T1 and T2 mapping techniques may identify the presence of early myocardial damage before the onset of myocardial fibrosis. Given that the degree of myocardial infiltration is diffuse, it is recommended to assess the value of native T1 in all myocardial segments (Figure 18) [77]. Native T1 time has been inversely correlated with wall thickness, so that, the more hypertrophy the lower T1. Additionally, patients with ECG abnormalities present shorter T1 [77]. In the early stages, ECV is normal, suggesting that LVH is due to myocyte hypertrophy and not to extracellular fibrosis.

FD must also be considered a chronic inflammatory disease. Thus, unlike patients with HCM, the presence of high native T2 values suggest FD, initially affecting the inferolateral wall and becoming progressively diffuse. Using 18F-fluorodeoxyglucose positron emission tomography (PET), an increase in the uptake of the tracer has also been described, confirming the presence of myocardial inflammation [78]. Finally, patients with FD also present a reduction in sympathetic activity, and 123I-meta-iodobenzylguanidine (MIBG) scintigraphy studies can differentiate stages of the disease [79]. The absence of myocardial denervation could play a relevant role in assessing the risk of developing ventricular arrhythmias and SCD.

### 4.4. Iron Overload Cardiomyopathy

Iron overload cardiomyopathy (IOC) is a secondary form of cardiomyopathy resulting from iron accumulation in the myocardium mainly because of genetically-determined iron metabolism disorders or multiple transfusions [80]. It has been described as a dilated cardiomyopathy, characterized by LV remodeling with chamber dilatation and reduced LVEF. However, primary hemochromatosis, a genetically determined condition leading to iron overload, is classically categorized as an infiltrative cause of restrictive cardiomyopathy. Moreover, secondary hemochromatosis may lead to severe diastolic LV dysfunction in the early stages of the disease, before LVEF is affected [80].

IOC can be very difficult to diagnose by TTE, therefore, CMR T2* imaging is considered the reference standard for detecting and quantifying myocardial iron overload. Abnormalities in CMR T2* can occur before the development of systolic or diastolic dysfunction and may be used to guide iron chelation therapy to prevent heart failure or death [81].

For easy classification into different IOC risk groups, the following classification has been adopted using a 1.5 Tesla MR scanner: patients with T2* >20 ms are regarded as not having cardiac iron overload, between 10–20 ms have mild to moderate cardiac iron load and those <10 ms are considered to have a heavy cardiac iron load [82].

### 4.5. Cardiac Sarcoidosis

Sarcoidosis is an inflammatory granulomatous disease that can involve any organ, with cardiac involvement (cardiac sarcoidosis, CS) in a quarter of patients. Clinical manifestations include heart block, atrial and ventricular arrhythmias, and heart failure. Diagnosis can be challenging but with the increased availability of advanced cardiac imaging, more cases are being identified.

TTE has limited sensitivity and specificity for the diagnosis of CS, however, it is often the initial imaging study. The TTE main findings include ventricular hypertrophy, diastolic dysfunction/restrictive filling pattern, wall motion abnormalities with a non-coronary distribution, aneurysms, and LV or RV systolic dysfunction [83]. CS patients who need pacing and have an LVEF <50% should be considered for CRT with ICD according to the recent ESC guidelines [84].

CMR plays an important role in the diagnosis and risk stratification of patients with CS. Although the presence of LGE may be non-specific, the subepicardial location, multifocal distribution, high signal intensity, and contiguous extension from the left to the right ventricle may increase the specificity of this finding for the diagnosis of CS [85]. Many studies have also demonstrated its prognostic value; LGE is associated with an increased risk of ventricular arrhythmias and all-cause mortality [85]. 

Cardiac PET using 18F-FDG has emerged as a cornerstone in the clinical diagnosis, prognostic evaluation, and monitoring of therapy in CS. FDG-PET/CT has a fair diagnostic accuracy for CS [86]. The classic pattern is one of ‘perfusion–metabolism’ mismatch, in which areas of 18F-FDG uptake correspond to areas of reduced or absent perfusion. FDG-PET/CT has a fair diagnostic accuracy for CS with a sensitivity of 89% and specificity of 78% [86]. An abnormal FDG uptake is associated with increased rates of ventricular arrhythmias and death, especially when located in the right ventricle [87] (Figure 19). Serial PET imaging is useful in monitoring disease activity and response to immunosuppressive therapy. Additionally, hybrid CMR-PET imaging has shown incremental value in determining disease activity and pattern [88].

### 4.6. Endomyocardial Fibrosis

Endomyocardial fibrosis (EMF) is a rare form of RCM characterized by an abnormal thickening of the endocardium due to fibrous tissue deposit [89] secondary to infections (typically in the tropical regions), inflammation, or toxic agents among others. Echocardiographic findings include apical obliteration due to endocardial thickening, a small ventricular cavity, and a marked restrictive diastolic pattern. EMF may affect primarily the left ventricle, both left and right ventricles (in approximately half of the cases), or predominantly the right ventricle [90]. Apical thrombus is also a common finding and echocardiographic contrast may be used to differentiate them from the thickened myocardium (Figure 20).

CMR is the gold standard for EMF evaluation and specifically for localization, characterization, and quantification of fibrous tissue by LGE sequences. LGE strongly correlates with histopathological findings and its extension is associated with increased mortality risk [91]. CMR may also identify apical thrombus or calcifications.

## 5. Arrhythmogenic Cardiomyopathy

Arrhythmogenic cardiomyopathy (ACM) is an inherited heart muscle disorder predisposing to SCD, particularly in young patients and athletes. It is a cell-to-cell junction cardiomyopathy, typically caused by genetically-determined abnormalities of cardiac desmosomes. Pathological features include loss of myocytes and fibrofatty replacement of right or left ventricular myocardium. ACM diagnosis does not rely on a single gold standard test but is classically achieved using a scoring system, which encompasses familial and genetic factors, ECG abnormalities, arrhythmias, and structural/functional ventricular alterations [92]. The score was recently updated and simplified [93].

TTE is the initial diagnostic approach in ACM patients. A thorough RV assessment with dedicated RV planes is recommended if clinical suspicion is high (Figure 21). The presence of regional RV akinesia, dyskinesia, or aneurysms and either RV dilatation or RV systolic dysfunction are considered diagnostic criteria on the 2010 Task Force Criteria, with different cut-off points for TTE and CMR major or minor criteria (Table 7) [92]. These were changed slightly in the Padua criteria [93], where RV dilatation and systolic dysfunction were defined according to nomograms, with no specific cut-off points. 

LV involvement is identified in more than half of patients with ACM, and biventricular as well as predominant LV ACM forms may occur [94], which are associated with a worse prognosis [95]. Specific diagnostic criteria for LV ACM were recently proposed: global LV systolic dysfunction (either LVEF or GLS) with or without LV dilatation, or regional LV hypokinesia/akinesia of the LV free wall or septum are considered minor criteria [93]. 

Emerging TTE parameters in the evaluation of patients with suspected or established ACM include the measurement of tricuspid annular plane systolic excursion (TAPSE), RV basal diameter, GLS (RV and LV), mechanical dispersion (RV and LV), and the use of 3D TTE. In particular, RV GLS is affected in early ACM phases [96], TAPSE (as well as fractional area change) are associated with worse outcomes [97], and RV mechanical dispersion correlates with ventricular arrhythmias risk [98]. Besides, LVEF has an incremental prognostic role over RV systolic function [99]. Altogether, an international consensus proposed that prophylactic ICD implantation should be indicated in case of severe RV/LV systolic dysfunction and should be considered if there is moderate RV or LV impairment [100].

Due to the limitations of TTE in RV evaluation, CMR has become an integral part of the diagnostic evaluation in ACM. Beyond determining the presence of morpho-functional ventricular abnormalities, CMR provides information on the presence, morphology, and wall distribution of myocardial fibrofatty scar by LGE or fat-saturation T1 sequences. The same criteria previously described for TTE apply to CMR with corresponding cut-off points and nomograms (Table 7) [92,93]. RVEF by CMR has been incorporated into the risk prediction model of ventricular arrhythmias [101]. The presence of transmural LGE of ≥1 RV region or LV LGE of the free wall (subepicardial or intramyocardial) or septum are considered major criteria [93]. Typically, the LGE pattern shows large amounts of contrast uptake in the LV with a non-ischaemic pattern, predominantly involving the subepicardial layers of the inferior and the inferolateral regions (Figure 22). The presence of a subepicardial annular (ring-like pattern) is also suggestive of ACM.

In cases where CMR is contraindicated, CT can be an alternative for the evaluation of RV and LV volumes and EF, aneurysms, fibrofatty infiltration, and wall motion abnormalities. Although generally not used in clinical practice for this purpose, angiography may be an alternative in the evaluation of these patients. 

## 6. Left Ventricular Noncompaction

Left ventricular noncompaction (LVNC) is a heterogeneous entity characterized by prominent LV trabeculae, deep intertrabecular recesses, and a thin compacted (C) myocardial layer [102] (Figure 23). Hypertrabeculation may occur associated with LV dilatation (DCM) or hypertrophy (HCM), and both acquired and genetic LVNC forms may occur. 

Different diagnostic criteria have been described for LVNC (Figure 24 and Figure 25). By TTE, the distance from the epicardial surface to the trough of the trabeculae (X) and the distance from the epicardial surface to the peak of the trabeculae (Y) can be measured on short-axis views. A ratio of X/Y ≤0.5 at end-diastole is diagnostic of LVNC [103]. Alternatively, a ratio of NC/C layers >2 measured on the short-axis at end-systole is also suggestive of LVNC [104]. By CMR, a ratio of NC/C layers >2.3 measured on long-axis views [105], a trabeculated mass >20% of the total LV mass [106], or a fractal dimension >1.30 [107] are diagnostic of LVNC (all measured at end-diastole). Fulfillment of LVNC morphologic criteria per se has not been associated with LV remodeling [108] or clinical events [109] throughout follow-up. However, the extension of the trabeculae has been recently related to outcomes: patients presenting hypertrabeculation from the apex to the base have been found to have higher mortality [110].

TTE is the first-line imaging technique for both quantification and localization of the trabeculae: the apex and lateral segments are the most frequently involved. Contrast echocardiography may be used to enhance trabeculation measurement and exclude the presence of intertrabecular thrombi (Figure 24B). Assessment of LVEF is mandatory since it is one of the strongest predictors of outcomes [110,111,112]. Strain analysis by TTE may detect subclinical myocardial dysfunction [113] and different studies have shown the diagnostic value of strain to differentiate LVNC from healthy controls [113] and DCM [114]. However, no consistent data is available on its prognostic implications.

CMR should be performed in all LVNC patients. The presence of LV dilatation [111,115] and a thinned compacted myocardial layer [115] has been associated with negative outcomes. Although LGE seems to be less frequent in LVNC compared to DCM or HCM [111], and not always related to the most hypertrabeculated segments, it is a powerful and independent predictor of outcomes, with added prognostic implications over LVEF [111,112,116]. Of note, patients with preserved LVEF and negative LGE have an excellent prognosis, while those with positive LGE are at increased risk irrespective of LVEF [116]. LGE has also been associated with SCD risk, even in the absence of severe systolic dysfunction [112]. Mapping techniques have been studied in LVNC; a small report has described worse outcomes with higher ECV values [117]. In addition, feature-tracking strain analysis may be used to differentiate LVNC from the general population [118] and DCM [119]. However, it is not recommended in routine practice due to lack of standardization and unvalidated association with clinical events.

Finally, cardiac CT is rarely used in LVNC, with its main objective being to exclude coronary artery disease. However, a large population study has demonstrated that the degree of hypertrabeculation measured by CT was independently related to outcomes [120]. Therefore, cardiac CT may be considered in patients with contraindications for CMR.

## 7. Conclusions

Cardiac imaging techniques are an essential tool in the study of cardiomyopathies, and they provide both diagnostic and prognostic relevant information. Transthoracic echocardiography should always be the first technique used due to its availability and cost, as well as the possibility of periodic testing. Cardiovascular magnetic resonance should also be performed in all patients due to its additional anatomical and functional information as well as thorough tissue characterization, with important prognostic implications. Other techniques (e.g., CT, PET-CT, scintigraphy, among others) may be applied in selected entities while more advanced imaging analysis (e.g., strain, myocardial work, mapping, etc.) offer promising but still preliminary results. Altogether, multimodality cardiac imaging plays an important role in clinical decision-making and helps to improve patients’ management and outcomes.

## Figures and Tables

**Figure 1 jcm-11-00578-f001:**
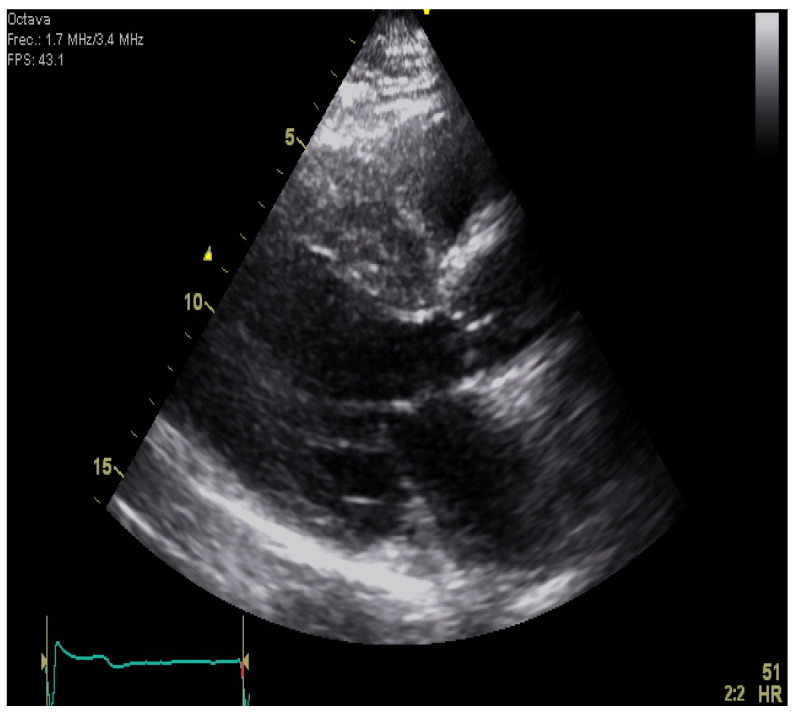
A parasternal long-axis view of a patient with septal HCM. Note the marked increase in septal wall thickness and the asymmetry compared to the posterior wall.

**Figure 2 jcm-11-00578-f002:**
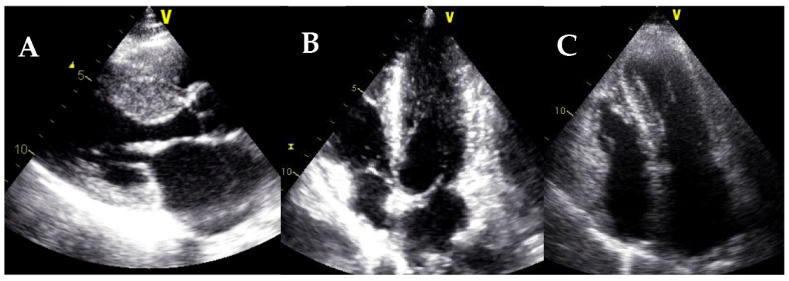
Different hypertrophic cardiomyopathy phenotypes. (**A**) Septal HCM. (**B**) Apical HCM. (**C**) Diffuse HCM.

**Figure 3 jcm-11-00578-f003:**
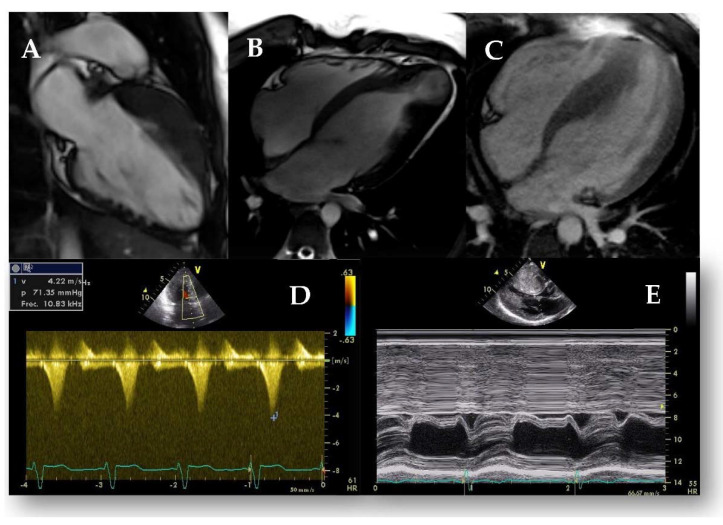
Ancillary signs in HCM. (**A**) Myocardial crypts in the inferior wall. (**B**) Apical aneurysm in a patient with apical HCM. (**C**) Apical insertion of the papillary muscles in a patient with septal HCM. (**D**) Outflow tract obstruction measured on Doppler CW, note the dagger-shaped curve. (**E**) SAM evidenced on M-mode: see the movement of the anterior mitral leaflet towards the septum in systole.

**Figure 4 jcm-11-00578-f004:**
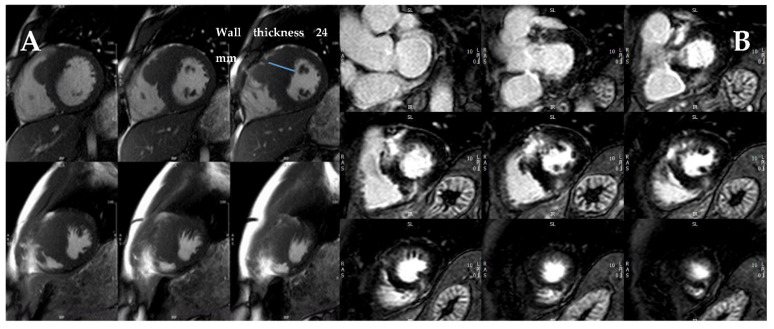
CMR study of a patient with septal HCM. (**A**) Maximal wall thickness measured on the short-axis cine stack at end-diastole. (**B**) Post-contrast T1 sequences showing extensive basal and mid-anteroseptal LGE with an intramyocardial pattern.

**Figure 5 jcm-11-00578-f005:**
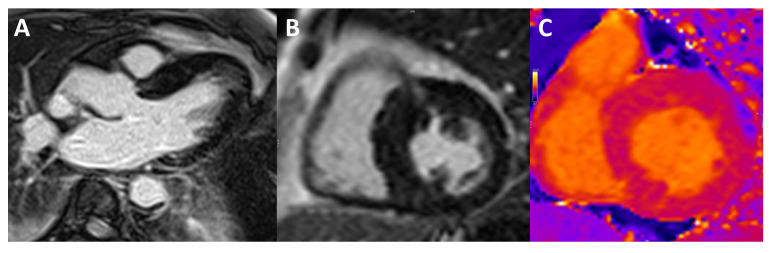
CMR study of a patient with septal HCM. (**A**,**B**) LGE sequences with heterogeneous contrast uptake in the basal and mid-anteroseptal segments. (**C**) Native T1 mapping sequence with an increased native T1 value of 1108 ms in the basal anteroseptal segment (reference range 950–1050 ms).

**Figure 6 jcm-11-00578-f006:**
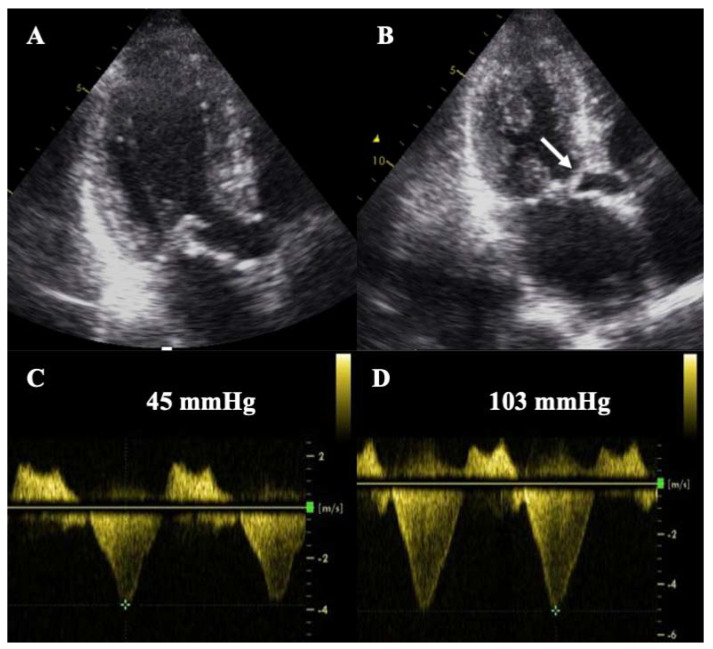
Stress echocardiography in a patient with HCM. Note the marked increase in SAM (**B**, white arrow) and the significant increase in LVOTO (**D**) at peak stress compared to the basal situation (**A**,**C**).

**Figure 7 jcm-11-00578-f007:**
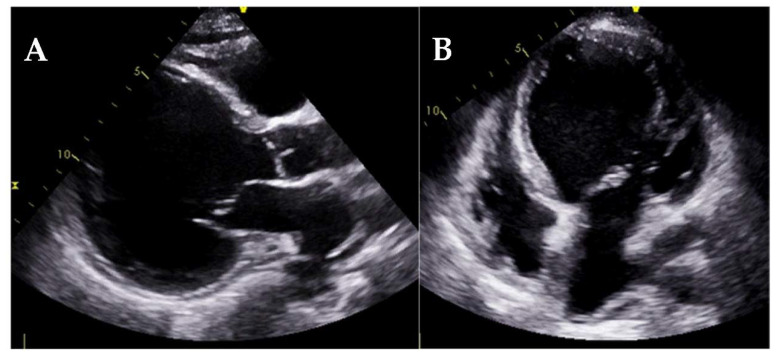
Transthoracic echocardiography of a patient with dilated cardiomyopathy, showing a parasternal long-axis view (**A**) and an apical 4-chambers view (**B**). Note the marked dilatation of the left ventricle and the spherical pattern.

**Figure 8 jcm-11-00578-f008:**
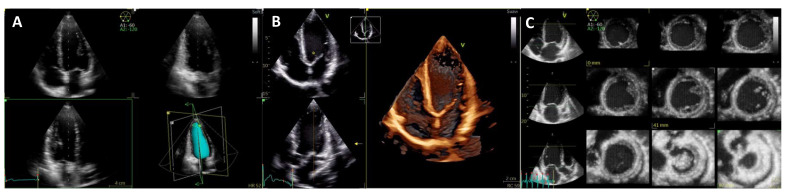
Three-dimensional transthoracic echocardiography of the left ventricle. (**A**) Triplane view of the LV allows simultaneous and single-beat acquisition of the three apical views. By tracing the endocardial borders, the LV volume is obtained (surface rendering, bottom right). (**B**) Real-time single-beat 3D acquisition of the LV from the apical window. Volume renders allow for offline reconstructions. (**C**) Tomographic multislice obtained from a multiple-beat apical view. Wall motion abnormalities can be assessed with this technique.

**Figure 9 jcm-11-00578-f009:**
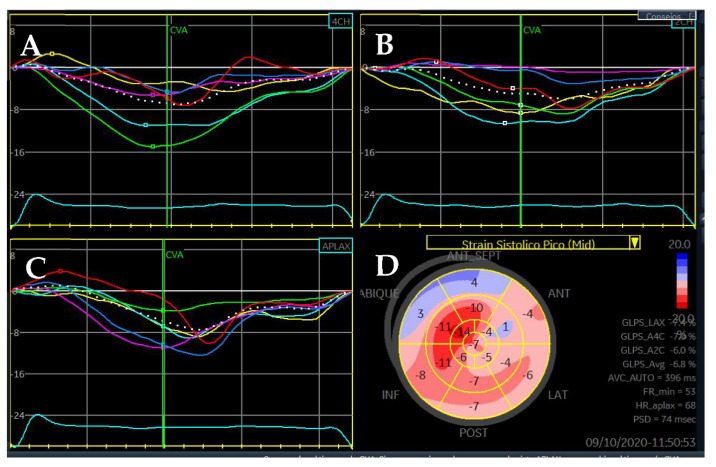
Longitudinal strain analysis of a patient with DCM: 4 chambers (**A**), 2 chambers (**B**), 3 chambers (**C**), and bull’s eye plot (**D**). Note the diffusely affected longitudinal strain, consistent with depressed LVEF, and positive values in the basal septum suggestive of dyskinesia in these segments.

**Figure 10 jcm-11-00578-f010:**
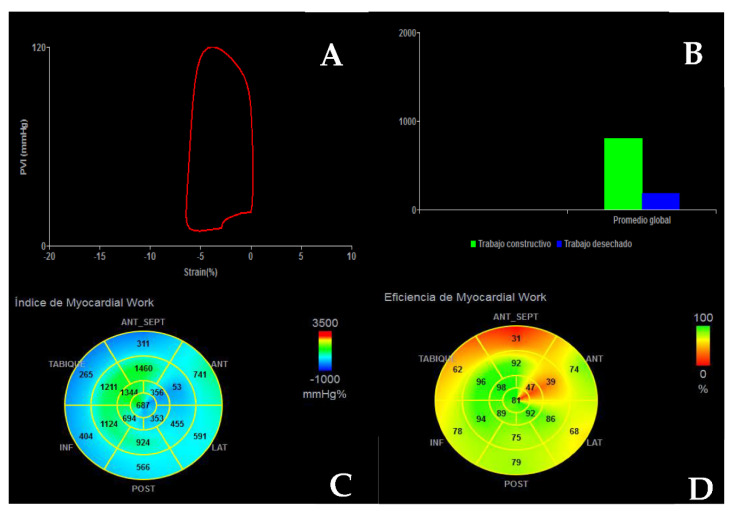
Myocardial work analysis in a patient with DCM. (**A**) Strain—pressure loop. (**B**) Comparison of constructive work (green) and wasted work (blue). (**C**) Bull’s eye plot of myocardial work index (mmHg%). (**D**) Bull’s eye plot of myocardial work efficiency (%).

**Figure 11 jcm-11-00578-f011:**
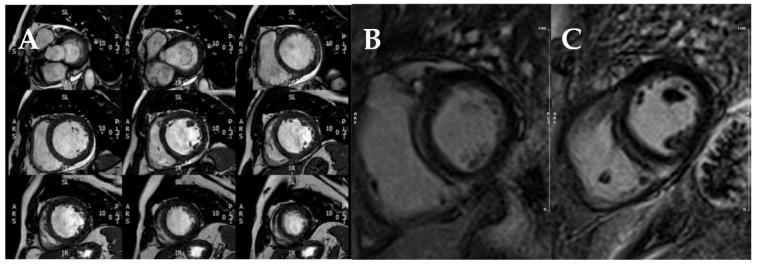
CMR study of a patient with DCM. (**A**) Short-axis cine stack. (**B**,**C**) Post-contrast T1 sequences showing a typical LGE pattern with a lineal mid-septum uptake as well as a focal uptake in the septum-free wall inferior junction.

**Figure 12 jcm-11-00578-f012:**
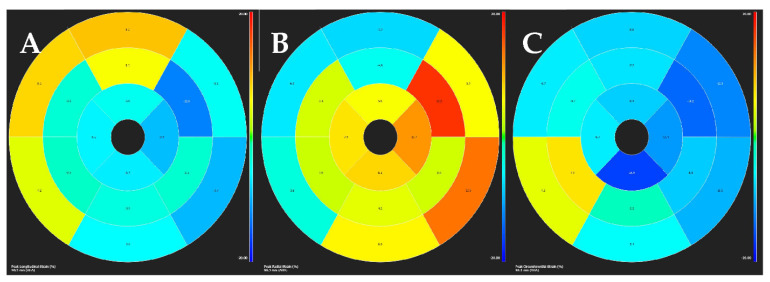
Strain analysis by CMR obtained with feature-tracking software in a patient with DCM. Endocardial and epicardial segmentation of short and long-axis is required, both at end-diastole and end-systole. Thus, longitudinal (**A**), radial (**B**), and circumferential (**C**) strain are simultaneously acquired.

**Figure 13 jcm-11-00578-f013:**
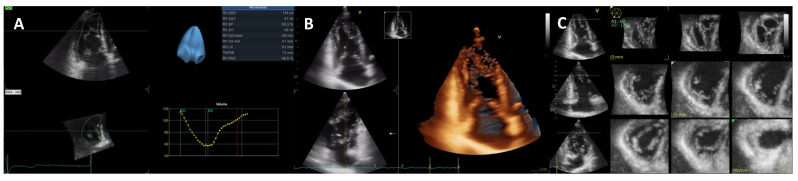
Three-dimensional transthoracic echocardiography of the right ventricle. (**A**) By tracing the endocardial RV borders (top and bottom left), a 3D volume of the RV (top right) is obtained throughout the cardiac cycle (bottom right) and 3D RVEF is calculated. Note the RV inflow and outflow tract in the 3D model. (**B**) Real-time single-beat 3D acquisition of the RV from the dedicated apical window. (**C**) Tomographic multislice obtained from a multiple-beat apical view.

**Figure 14 jcm-11-00578-f014:**
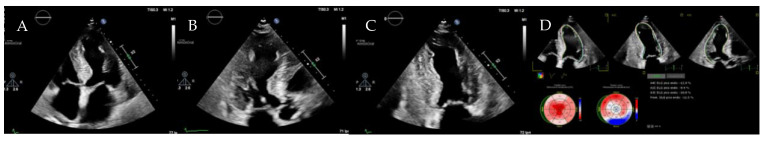
Apical 4-chamber (**A**), 3-chamber (**B**), and 2-chamber (**C**) views in a patient with cardiac amyloidosis (left ventricular wall thickness and sparkling). On the bull’s eye strain images (**D**), the apical sparing pattern is displayed (right image).

**Figure 15 jcm-11-00578-f015:**
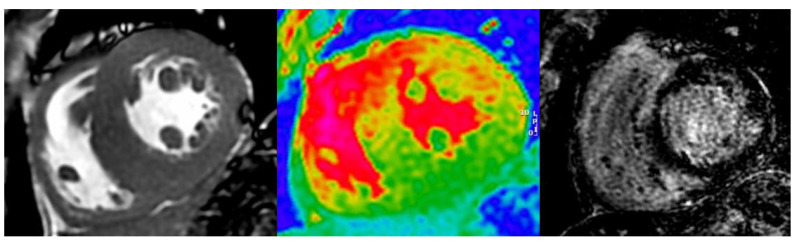
CMR findings in a patient with CA: left ventricular concentric hypertrophy (left), increased native T1 mapping values (red color in the central image), and diffuse LGE (right image).

**Figure 16 jcm-11-00578-f016:**
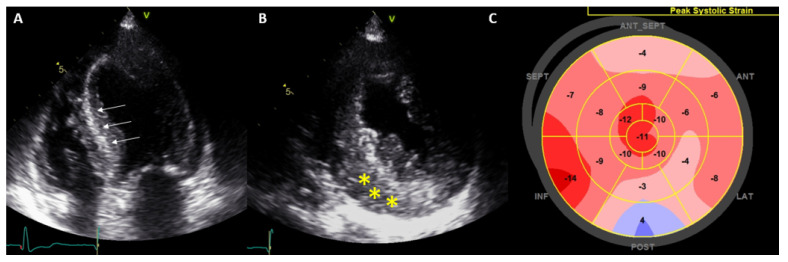
Echocardiographic signs in Fabry disease. (**A**) Binary septum (arrows), (**B**) concentric hypertrophy, and inferolateral fibrosis (*), (**C**) abnormal longitudinal strain more pronounced in the inferolateral wall (blue area) correlated with the fibrosis.

**Figure 17 jcm-11-00578-f017:**
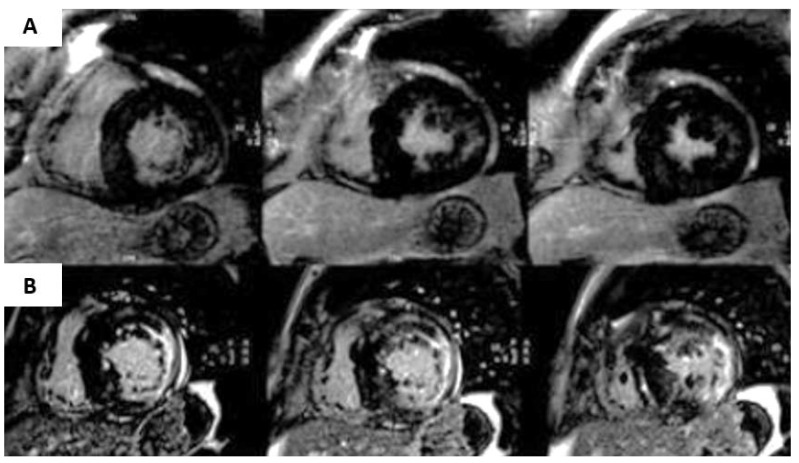
Late gadolinium enhancement pattern in a patient with Fabry disease. (**A**) Presence of LGE in the inferolateral wall. (**B**) Evolution of the same patient 5 years later, the presence of a thinning of the wall, a more extensive and transmural LGE pattern is observed.

**Figure 18 jcm-11-00578-f018:**
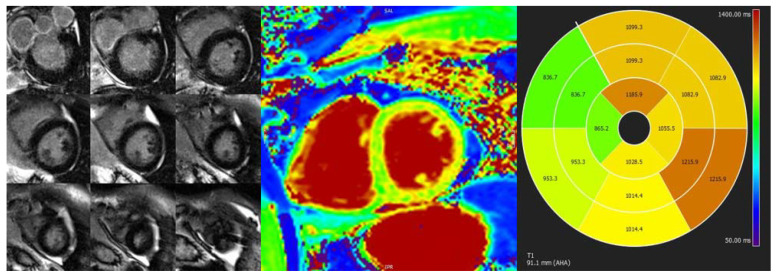
CMR findings in a patient with Fabry disease: (Left panel) presence of concentric left ventricular hypertrophy with no LGE. (Central panel) T1 mapping image of the same patient. (Right panel) bull’s eye representation of the T1 mapping showing a short native T1 in the septum (glycosphingolipid accumulation) and a long T1 value in the inferolateral wall (diffuse fibrosis). T1 mapping abnormalities appear earlier than LGE.

**Figure 19 jcm-11-00578-f019:**
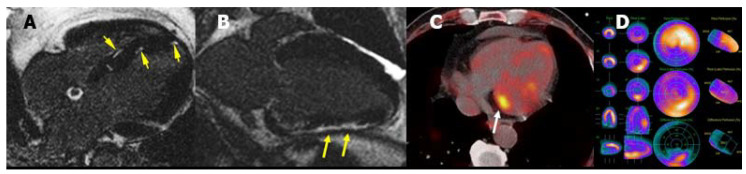
(**A**) Patchy LGE distribution in the septum (arrows) and (**B**) subepicardial LGE (arrows) in a patient with sarcoidosis. (**C**) 18F-FDG uptake in a patient with sarcoidosis (arrow). (**D**) Perfusion-metabolism mismatch in a patient with sarcoidosis (perfusion defect in the upper image, metabolism uptake in the mid image, and mismatch in the lower image).

**Figure 20 jcm-11-00578-f020:**
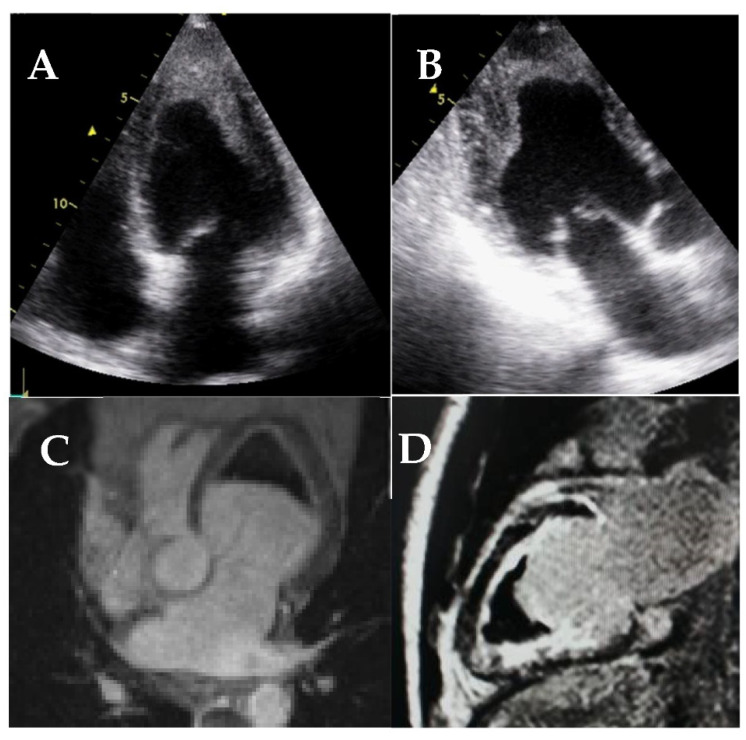
Transthoracic echocardiography (**A**,**B**) and cardiac magnetic resonance (**C**,**D**) of patients with endomyocardial fibrosis. Note the marked endocardial thickening of the mid and apical segments and apical obliteration of the left ventricle (**A**,**B**), the apical thrombus (**C**,**D**), and the endomyocardial fibrosis on LGE sequences (**D**).

**Figure 21 jcm-11-00578-f021:**
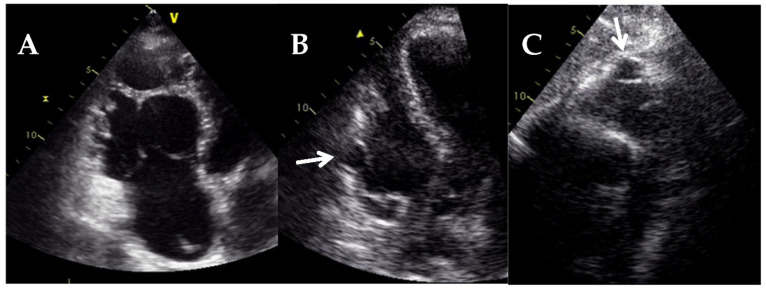
Transthoracic echocardiography of a patient with right ventricular arrhythmogenic cardiomyopathy. (**A**) RV dedicated apical 4-chamber view shows severe dilatation of the RV. RV dedicated apical 4-chamber view (**B**) and subcostal view (**C**) show the presence of aneurysms in the RV free wall (arrow).

**Figure 22 jcm-11-00578-f022:**
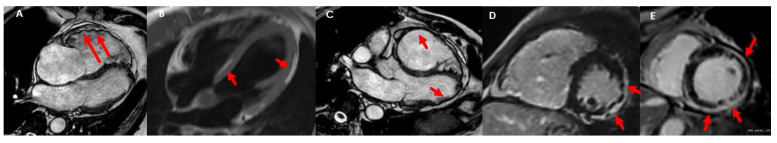
CMR findings in a patient with ACM. (**A**) RV aneurysms (arrows) in cine images. (**B**) Fibrofatty infiltration in T1-weighted turbo spin-echo sequences (arrow). (**C**) Right ventricular enlargement and left ventricular wall thinning (subepicardial fatty infiltration) (arrows) in a 3-chamber view cine. (**D**) Subepicardial LGE (arrows) and (**E**) subepicardial annular (arrows) patterns.

**Figure 23 jcm-11-00578-f023:**
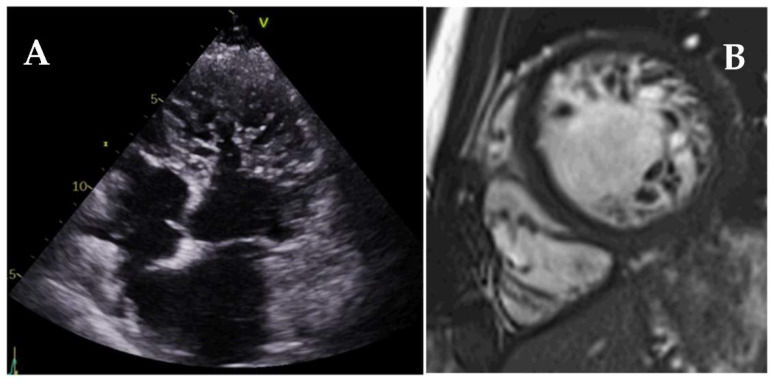
TTE (**A**) and CMR (**B**) images of patients with left ventricular noncompaction showing marked hypertrabeculation and deep intertrabecular recesses.

**Figure 24 jcm-11-00578-f024:**
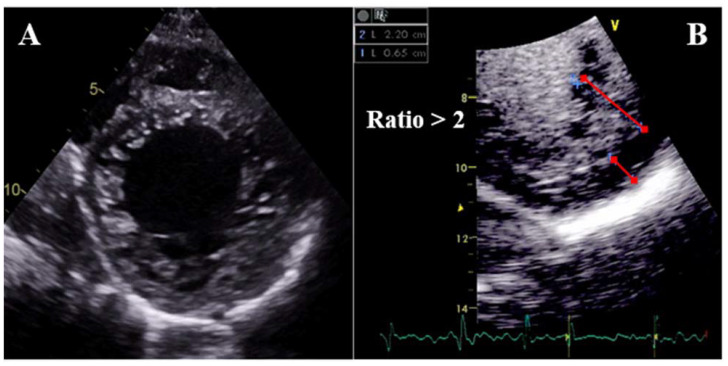
TTE parasternal short-axis mid-ventricular view of a patient with LVNC (**A**). Measurement of the compacted and non-compacted layers at end-diastole after echocontrast administration, fulfilling LVNC diagnostic criteria (**B**).

**Figure 25 jcm-11-00578-f025:**
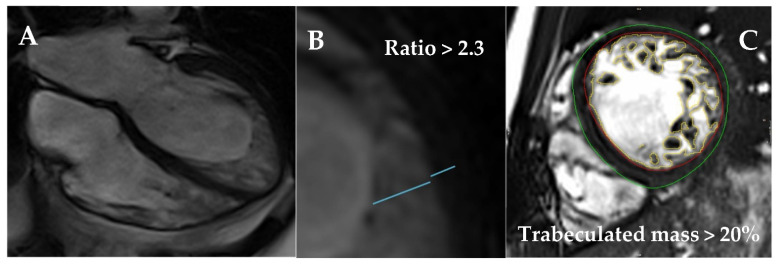
CMR 4-chambers view of a patient with LVNC (**A**). Measurement of the compacted and non-compacted layers at end-diastole on a long-axis view (Petersen criteria, **B**). Measurement of the trabeculated mass on a short-axis view (Jacquier criteria, **C**).

**Table 1 jcm-11-00578-t001:** Summary of imaging findings in HCM.

Diagnostic Criteria	Ancillary Signs	Prognostic Markers
Maximal wall thickness >15 mm (>13 mm in relatives of HCM patients)	Papillary muscle abnormalities	Maximal wall thickness (>30 mm)
Asymmetric septal hypertrophy (ratio septal/posterior wall thickness >1.3 or >1.5 in hypertensive patients)	Mitral valve and subvalvular structure abnormalities	Left atrial size (anteroposterior diameter)
	Myocardial clefts/crypts	Maximum LVOT gradient (at rest or induced by exercise)
	Aneurysms	LVEF < 50%
		Abnormal GLS
		Presence, extension, and progression of LGE
		Increased T1/ECV values

**Table 2 jcm-11-00578-t002:** Summary of prognostic imaging markers in DCM.

LVEF ≤ 35%
RV systolic dysfunction (RVEF < 45%)
Significant (secondary) mitral regurgitation
Advanced diastolic dysfunction
Abnormal strain and myocardial work values
Presence, extension, pattern, and progression of LGE
Increased T1 and ECV values

**Table 3 jcm-11-00578-t003:** Classification of restrictive cardiomyopathies.

Restrictive Cardiomyopathy			
Non-Infiltrative Disorders	Infiltrative Disorders	Storage Diseases	Endomyocardial Diseases
Idiopathic Hereditary (sarcomere mutations…) Systemic sclerosis	Amyloidosis Sarcoidosis Hereditary hyperoxaluria	Anderson–Fabry disease Danon disease Pompe disease Gaucher disease Iron overload Hereditary hemochromatosis	Carcinoid Endomyocardial fibrosis Endocardial fibroelastosis Metastatic tumor Chemotherapy Radiation therapy

**Table 4 jcm-11-00578-t004:** Main differences between ATTR and AL amyloidosis.

	ATTR	AL
Left ventricular wall thickness and LV mass	++++	++
Asymmetrical septal hypertrophy	79%	14%
Transmural LGE	63%	27%
Subendocardial LGE	24%	39%
Native T1 elevation	++	++++
Native T2 relaxation time	++	++++
ECV	++++	++

++ Mildly abnormal. ++++ Severely abnormal.

**Table 5 jcm-11-00578-t005:** Main imaging red flags in cardiac amyloidosis.

Echocardiography	CMR	Scintigraphy
Left ventricular wall thickness > 12 mm. Myocardial sparkling Pleural/pericardial effusion Increased valvular thickness Thick interatrial septum. Low stroke volume. Paradoxical low-flow low-gradient aortic stenosis. Restrictive filling pattern. Apical sparing pattern. Abnormal left atrial strain.	Increased LV wall thickness. Increased myocardial LV mass Bi-auricular enlargement Difficulty to null the myocardial signal Global or diffuse LGE, Marked increase in native T1 values Elevated extracellular volume (> 40%) Presence of pleural or pericardial effusion	Grade 2 or 3 uptakes in the Perugini scale. C/CL uptake ≥ 1.5

**Table 6 jcm-11-00578-t006:** Main echocardiographic findings in Fabry disease.

CARDIAC STRUCTURE	FINDINGS
Left ventricle	Concentric left ventricular hypertrophy Binary septum (low sensitivity and specificity for FD) Prominent papillary muscles Preserved LV function until end-stages Diastolic dysfunction Abnormal global longitudinal or radial strain, even in the absence of LVH
Right ventricle	Right ventricular hypertrophy Preserved RV function until end-stages Abnormal global longitudinal strain even with preserved EF
Atrium	Biauricular dilation Increased end-diastolic pressure Reduced auricular strain
Valves	Increase in mitral and aortic valve thickness Valvular regurgitation (generally, mild)
Aorta	Dilation of the aortic root and the ascending aorta (not descending aorta)

**Table 7 jcm-11-00578-t007:** 2010 Task Force Criteria for the Diagnosis of Arrhythmogenic Right Ventricular Cardiomyopathy. Adapted from [92].

Global and/or Regional Dysfunction and Structural Alterations	
Major	By 2D TTE: regional RV akinesia, dyskinesia, or aneurysm and 1 of the following (end diastole): PLAX RVOT ≥ 32 mm (corrected for body size [PLAX/BSA] ≥ 19 mm/m^2^) PSAX RVOT≥ 36 mm (corrected for body size [PSAX/BSA] ≥ 21 mm/m^2^) Or fractional area change ≤ 33%
	By CMR: regional RV akinesia or dyskinesia or dyssynchronous RV contraction and 1 of the following: Ratio of RV end-diastolic volume to BSA ≥ 110 mL/m^2^ (male) or ≥100 mL/m^2^ (female) Or RV ejection fraction ≤ 40%
	By RV angiography: regional RV akinesia, dyskinesia, or aneurysm
Minor	By 2D-TTE: regional RV akinesia or dyskinesia and 1 of the following (end-diastole): PLAX RVOT ≥ 29 to <32 mm (corrected for body size [PLAX/BSA] ≥ 16 to <19 mm/m^2^) PSAX RVOT ≥ 32 to <36 mm (corrected for body size [PSAX/BSA] ≥ 18 to <21 mm/m^2^) Or fractional area change >33% to ≤40%
	By CMR: regional RV akinesia or dyskinesia or dyssynchronous RV contraction and 1 of the following Ratio of RV end-diastolic volume to BSA ≥100 to <110 mL/m^2^ (male) or ≥90 to <100 mL/m^2^ (female) Or RV ejection fraction >40% to ≤45%

## Data Availability

See references.

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
