# Peer review of "Multimodality Cardiac Imaging in Cardiomyopathies: From Diagnosis to Prognosis"

_jcm, 2022, doi:10.3390/jcm11030578_

Round 1

Reviewer 1 Report

Dear authors,

it is a good review of the imaging techniques in different cardiomyopathies, focused on the information obtained using different cardiac imaging techniques and what they can offer to clinicians. 

I miss the role of stress echocardiography, often used in clinical practice for diagnosis, risk stratification, and assessment of therapy response: especially exercise testing in hypertrophic cardiomyopathy (exertion-induced LVOTO, diastolic dysfunction, dynamic MR, inducible ischaemia) and dobutamine in dilated cardiomyopathy for contractile reserve.

I would appreciate it if the 3D echocardiography would be mentioned more, with some images included (it is mentioned just for ACM). It is an imaging technique - a complement to 2D echocardiography, present in echocardiography guidelines. 3DE is superior to 2DE and has a closer agreement with CMR in the evaluation of LV volumes and geometry of cardiac chambers. It is an evolving concept, there are increasing numbers of studies in the literature and still, more sophisticated software (fully automated) are available. So it is worth mentioning it separately from 2DE.

The message that the use of multimodality cardiac imaging is crucial for the patient is clear. In the literature, we usually find the articles divided by techniques, not by cardiomyopathies like in this article. Maybe even Takotsubo cardiomyopathy could be included in "unclassified cardiomyopathies", not only LVNC. Let me allow to point out that there is a little mistake in paragraph IV: the subgroups of restrictive cardiomyopathy are marked as III.1, III.2 etc instead of IV.1, IV.2 etc.

Author Response

Dear reviewer, 

Thank you for your appropriate comments. They have all been addressed and have contributed to improving the manuscript quality.

New paragraphs about stress echocardiography have been added, both in HCM and DCM, together with a new Figure. The role of stress echocardiography in risk stratification and patient management has also been comented. This is an important technique in clinical practice and the mauscript is now more robust.

A new paragraph about 3D echocardiography has been added. The advantages and disadvantages of 3DE have been comented, both for LV and RV assessment. New composite images of 3DE have been added for LV and RV. We hope the text and images are illustrative of this important technique, which was an important absence of the previous version of the manuscript.

Due to the limit of words, Takotsubo cardiomyopathy has not been added to the manuscript. In addition, it's not strictly an inherited cardiomyopathy. Finally, the titles in section IV have been changed.

Thank you again for you comments. We think that the updated version is substantially better.

Best regards.

Reviewer 2 Report

This article discusses the main structural cardiological pathologies and the means to perform adequate diagnosis and therapy.
Almost all the diseases analyzed require transthoracic echocardiography to be classified initially, and then second-level examinations are performed to better characterize their stage and prognosis.
Despite some inaccuracies in the use of the English lexicon, the arguments expressed in this article are important for the classification of all the pathologies under examination; it is important to understand which second-level examinations should be performed for the staging of the disease and the subsequent treatment.
I suggest that the following article be consulted and, if possible, quoted:
PMID: 32169347

Author Response

Dear reviewer,

We would like to sincerely thank you for your comments. We apologize for the English accuracy, the manuscript has been revised by a native English speaker. 

In addition, the appropriate reference you suggested has now been added.

We hope you find the new version adecuate.

Best regards.